# Susac Syndrome: Description of a Single-Centre Case Series

**DOI:** 10.3390/jcm11216549

**Published:** 2022-11-04

**Authors:** Sara Beça, Claudia Elera-Fitzcarrald, Albert Saiz, Sara Llufriu, Maria C. Cid, Bernardo Sanchez-Dalmau, Alfredo Adan, Gerard Espinosa

**Affiliations:** 1Department of Autoimmune Diseases, Institut d’Investigacions Biomèdiques August Pi i Sunyer (IDIBAPS), Hospital Clinic, 08036 Barcelona, Spain; 2Professional School of Human Medicine, Private University of San Juan Bautista, Lima 15067, Peru; 3Center for Neuroimmunology, Department of Neurology, Institut d’Investigacions Biomèdiques August Pi i Sunyer (IDIBAPS), Hospital Clinic and Barcelona University, 08036 Barcelona, Spain; 4Department of Ophthalmology, Institut d’Investigacions Biomèdiques August Pi i Sunyer (IDIBAPS), Hospital Clinic, 08036 Barcelona, Spain

**Keywords:** Susac syndrome, branch retinal artery occlusion, central nervous system disorder, sensorineural hearing loss

## Abstract

This study describes the clinical characteristics, diagnostic results, treatment regimens, and clinical course of a cohort of patients with Susac syndrome (SS). It is a retrospective observational study of all patients with the diagnosis of SS evaluated at the Hospital Clinic (Barcelona, Spain) between March 2006 and November 2020. Nine patients were diagnosed with SS. The median time from the onset of the symptoms to diagnosis was five months (IQR 9.0), and the median follow-up time was 44 months (IQR 63.5). There was no clear predominance of sex, and mean age of symptoms onset was 36 years (range 19–59). Six patients (67%) presented with incomplete classical clinical triad, but this eventually developed in six patients during the disease course. Encephalopathy, focal neurological signs, visual disturbances, and hearing loss were the most frequent manifestations. Brain magnetic resonance imaging showed callosal lesions in all patients. Most were in remission within two years. Only four patients met the proposed criteria for definite SS. When SS is suspected, a detailed diagnostic workup should be performed and repeated over time to identify the clinical manifestations that will lead to a definite diagnosis.

## 1. Introduction

Susac syndrome (SS) is a rare disease characterized by a clinical triad of encephalopathy, sensorineural hearing loss, and visual disturbance [1]. Its aetiology is unknown, but it is presumably an autoimmune inflammatory and probably T-cell mediated endotheliopathy that results in arterial occlusions involving the brain, ear, and retinal vessels [2,3,4,5,6]. The incidence is higher in females than in males with a reported ratio of 3.5:1 [7]. Owing to its rarity, prospective or randomised controlled treatment trials do not exist, and therapeutic approaches in the acute phase have included high-dose glucocorticoids, immunosuppressive agents, intravenous immunoglobulins (IVIG), tumour necrosis factor inhibitors, plasma exchange, rituximab, antiplatelet agents, and anticoagulation therapy [7,8,9,10,11]. Unfortunately, irreversible damage of neurological, auditory, and/or ocular systems is frequent [7]. The purpose of the present study is to describe the demographics, clinical characteristics, imaging findings, treatments, and outcomes of nine patients with SS attended in a Spanish centre.

## 2. Materials and Methods

A retrospective medical record review was performed on all patients with the diagnosis of SS evaluated at the Hospital Clinic, Barcelona from March 2006 to November 2020. In all the patients, a complete diagnostic work-up was implemented to exclude other aetiologies, such as inflammatory demyelinating central nervous system (CNS) disease, cerebrovascular disease, systemic autoimmune diseases, and primary vasculitis, infections, and malignancies.

The following variables were collected from the medical records of each patient: (a) demographics (sex and age at symptoms onset and diagnosis); (b) clinical features at onset and during the disease course, including neurological, ocular, and hearing characteristics; (c) diagnostic procedures at presentation or during follow-up, including brain magnetic resonance imaging (MRI), audiometry, retinal fluorescein angiography (FA), optical coherence tomography angiography (OCT-A), cerebrospinal fluid (CSF) analysis, electroencephalogram, and autoantibodies determination. The clinical course of the disease was stratified according to Rennebohm et al. [12] into three forms: monocyclic (remission in one to two years, without recurrence); polycyclic (patients experience recurrences alternating with remission periods in a time frame superior to two years); and chronic–continuous (disease continuously active, usually to fluctuating degrees, for more than two years). Data on treatment, follow-up, and outcome focusing on organ damage were also collected. The European Susac Consortium (EuSaC) diagnostic criteria [13], the type of clinical course described by Rennebohm R et al. [12], and the modified Rankin scale score [14] were assigned retrospectively by the authors based on detailed clinical notes.

Concerning the statistical analysis, categorical data are summarised as percentages, and continuous variables are presented as mean ± standard deviation (SD) or median (interquartile range, IQR), depending on the normality demonstrated by the Shapiro–Wilk test.

The study was designed by the authors and approved by the Ethical Committee of the Hospital Clinic, Barcelona, which waived the requirement for individual informed consent.

## 3. Results

### 3.1. General Characteristics

This series includes nine patients diagnosed with SS. There was no clear predominance of sex; five patients were male and four were female. The mean age of symptoms onset was 36 ± 13.8 years (range 19–59 years) with a median of diagnosis delay of five months (IQR 9.0). The median follow-up was 44.0 months (IQR 63.5). Demographic features, clinical manifestations, and disease course of each patient are described in Table 1.

### 3.2. Onset of Susac Syndrome and Course of the Disease

The complete clinical triad was the initial presentation in three out of nine (33%) patients and developed later in three more (Table 1). At disease onset, the most common symptoms referred to the CNS, were present in eight (88%) patients, and ophthalmologic features were present in seven (78%), and ear manifestations in only five (56%) patients (Table 1). None of the patients reported infectious episodes antedating the disease presentation. In three patients, a toxic exposure to drugs (cocaine in all three) was identified. Five out of eight patients (with a follow-up longer than two years) had a monocyclic course. All men (except the one missing during the follow-up) had a monocyclic course. The three patients with relapsing courses had a mean number of relapses of 2.7 (range 1–4). The longest time between exacerbations of the disease was 93 months.

The Table 2 describes the organ involvement at presentation, during the disease course, and the persistent manifestations at the final visit.

The main results of diagnostic procedures, treatment, and prognosis are described in Table 3.

#### 3.2.1. Neurological Characteristics

All patients developed neurologic symptoms over time. Encephalopathy and focal neurological symptoms were the most frequent clinical features, occurring in seven out of nine patients each. Among the seven patients with encephalopathy, six had confusion/disorientation, five had cognitive impairment, three had consciousness impairment, and two presented behavioral changes. Ataxia (four patients), sensory symptoms (four patients), focal motor deficits (three patients), dysarthria and diplopia (two patients each) were the other recorded neurological manifestations. Four (44%) patients had residual neurologic symptoms at the end of follow-up.

Brain MRI was abnormal in all patients (Table 3). Hyperintense lesions located in the corpus callosum on fluid-attenuated inversion recovery (FLAIR) MRI were seen in all patients (Figure 1), followed by lesions in the periventricular area (eight patients, 89%), subcortical white matter and brainstem (four patients each, 44%), basal ganglia and cerebellum (three patients each, 33%). MRI lesions were multiple and bilateral in all cases.

CSF analysis was performed in eight patients (Table 2). CSF examination often showed an elevation of total protein (five out of eight patients, 62%) with mean levels of 117 mg/dL (range 29 to 238 mg/dL). Mild pleocytosis, defined as more than five white-blood cells in CSF, was observed in only three patients and was always ≤10 cells/μL. No CSF oligoclonal bands were documented in the six patients in whom they were examined.

#### 3.2.2. Ocular Characteristics

During the follow-up, eight (89%) patients had ophthalmologic manifestations. The main symptoms were scotoma and blurred vision. Branch retinal artery occlusion (BRAO), the most characteristic ophthalmologic lesion in retinal FA or OCT-A, was found in eight (89%) patients (Figure 2A,B). One patient had no ocular symptoms but presented BRAO on ophthalmological exam. In seven patients, ocular involvement was bilateral. In four cases (44%), signs of arteriolar vasculitis were also present (Figure 2C). At the end of follow-up, four out of the nine patients had ocular damage (Table 2).

#### 3.2.3. Ear Characteristics

Hearing loss of varying intensity was present in seven (78%) patients. Three patients presented with vertigo, and one referred to tinnitus. The nine patients underwent an audiometry that revealed sensorineural hearing loss in six (67%) and transmission hearing loss in one patient. These findings were reported bilaterally in four cases. Six patients maintained residual hearing loss at the end of follow-up (Table 3).

### 3.3. Diagnostic Criteria

Applying the diagnostic criteria proposed by the EuSaC [13] to our cohort at the time of diagnosis, four (44%) fulfilled the criteria for definite SS and five (56%) for probable disease. During the follow-up period, one patient was reclassified as definite SS.

### 3.4. Laboratory Markers

Five (56%) patients had low levels (≤1:80) of antinuclear antibodies (ANA). One patient had a positive test for IgM anti-ß2-glycoprotein I antibody and lupus anticoagulant, but both were negative in later determinations. Anti-neutrophil cytoplasm antibodies (ANCA) were negative in all patients.

### 3.5. Treatment

All patients received glucocorticoids, most of them in the form of pulses of methylprednisolone followed by high doses of oral prednisone (usually 1 mg/kg/day). As induction treatment, most patients received a combination of steroids and other immunosuppressive agents (cyclophosphamide or mycophenolate mofetil) or steroids and IVIG (Table 3). Two patients received the triple therapy of glucocorticoid, cyclophosphamide or mycophenolate mofetil, and IVIG as induction therapy. Treatment of the exacerbations included pulses of methylprednisolone, change of immunosuppressive agent, and/or addition of IVIG. Two patients received rituximab for refractory disease. Additionally, one patient received infliximab and plasmapheresis and was submitted to an autologous stem cell transplantation because of the severe and refractory disease. This last case was reported previously, and after five years of follow-up, the patient is in remission without any treatment [15].

Given the vasculopathy present in SS, all patients were treated with an antiplatelet agent as an add-on therapy. One patient received anticoagulation therapy because she had concomitant protein S deficiency.

### 3.6. Outcome

At the last visit, seven of the nine (78%) patients had variable degrees of organ damage. Most of them had sensorineural hearing deficit (six patients, 67%). CNS damage was present in four patients with variable manifestations, the most common being headache, motor deficit, and cognitive impairment. Residual visual loss was also present in four patients (Table 3).

Although most patients had some degree of organ damage at the end of the follow-up, the majority had no limitations in their usual activities with a median of modified Rankin Scale score of 1 (IQR 1.0). Of note, the patient with the highest modified Rankin scale score had physical limitations secondary to multiple vertebral fractures resulting from previous high doses of glucocorticoids and not directly related to the disease involvement.

## 4. Discussion

This case series describes nine patients with probable or definite SS, diagnosed and/or treated in a tertiary hospital in Spain over approximately fifteen years. The low number of patients over such a long time frame is consistent with the rarity of this diagnosis and suggests the disease may be underdiagnosed.

Unlike previous reports, in the present series there was no clear predominance of female sex [3,7,16]. The median of diagnosis delay was five months, comparable to a review of all published cases [7] but longer than the duration reported in recent series [9,16,17] (Table 4). This fact may indicate that SS is currently considered more often in the differential diagnosis and appropriate diagnostic tools used earlier.

In our series, three patients had histories of cocaine consumption. In this context, the rare diagnosis of multifocal inflammatory leukoencephalopathy or a toxic-induced vasculopathy should be considered. When these patients were compared with those without cocaine exposure, there were no differences regarding the clinical manifestations, results of diagnostic workup, or response to therapy, thus suggesting that the diagnosis of SS was accurate. Anecdotally, these patients had shorter times from symptoms onset to diagnosis, and two had behavioral changes not reported in the rest of cases. All had a monocyclic course of the disease, similar to the majority of our series, but limited information was detailed on medical records concerning the pattern of consumption or abstinence during follow-up. From another perspective, the potential contribution of toxic agents to the development of SS has not been extensively investigated, although some authors reporting single cases have suggested that cocaine use, particularly in the presence of adulteration by levamisole, could play a role as a triggering cofactor in the onset of SS or Susac-like syndromes [18,19]. However, a causal association between drug use and the appearance of SS cannot be established.

Overall, the clinical presentation and disease course of our cohort is similar to other published reports [3,7,8,9,11,16,17,20]. Compared with literature (Table 4), a higher percentage of patients presented with the classic triad at disease onset [3,7,16,17]. However, a correlation between the full triad presentation and a shorter interval from the first symptoms to the diagnosis could not be made. Neurological manifestations were the most common presentation at disease onset, particularly with encephalopathy and focal neurological symptoms. During the entire disease course, eye and ear manifestations developed in an equal proportion of patients. Most of the cases had at least one exacerbation of the disease. The natural history of a patient with SS is difficult to predict, but it is noticeable that all men in this series had a monocyclic course, which means that the disease remitted in two years and did not recur. But it is important to note that in one patient, the interval between the diagnosis and the first exacerbation was approximately eight years, which reflects the high variability of the course of the disease and advises about the possible underestimation of recurrence in patients with shorter follow-up. Although the clinical course of SS has been broadly characterized as monocyclic, polycyclic, or chronic continuous, the validity of these concepts was assessed by Vodopivec and Prasad [21], and they found that a short follow-up bias in reported cases confounds the true long-term outcome of the disease, considering that relapses have been observed after 10 years of remission. Importantly, the distinction between a relapse and unremitting low-grade disease activity may be challenging given the lack of accepted definitions for relapse, remission, or intermittent disease. Therefore, the stratification of SS into monocyclic, polycyclic, or chronic continuous forms may be inaccurate, and SS may be best regarded as a disease with a wide range of outcomes that may be altered by treatment.

The most typical findings in patients with SS, such as characteristic callosal lesions on cranial MRI, BRAO detectable on retinal FA or OCT-A, and evidence of sensorineural hearing loss on audiometry, were present in 100%, 89%, and 67% of patients, respectively. As already reported in literature, one patient had subclinical pathology (eye lesions without ocular symptoms), emphasising the importance of retinal FA and OCT-A to enable an accurate diagnosis even when the clinical triad is not present. With respect to immunological features, only low titers of ANA were observed (in around half of the patients), confirming that detection of elevated ANA is not relevant to the diagnosis of SS. All other immunological markers were negative. Anti-endothelial cell antibodies (AECA) were not consistently checked. Although high levels of AECA have been reported, titers > 1:100 were found in only 25% of patients with SS [3] and were also identified in the majority of autoimmune diseases and vasculitis [22]. Overall, these data are similar to those previously published [3,7,8], suggesting that SS does not have a known specific immunological marker, and the broad screening for autoantibodies is only necessary for differential diagnostic purposes.

Diagnostic criteria for SS have been proposed by the EuSaC [13]. According to these criteria, patients with involvement of all three main organs (brain, eye, and ear) who fulfill the typical clinical triad (considering both symptoms and clinical findings) were defined as definite SS, and patients with involvement of two main organs were defined as probable SS. Among the nine patients of the present series, four (44%) met the proposed criteria for definite SS at the time of diagnosis. The other five patients had probable SS due to incomplete triad in four cases, and in one case, despite the presence of symptoms from all the three mainly affected organs, the performed audiogram reported a transmission hearing loss without reference to a sensorineural component. During the follow-up, only one of these patients completed the triad and could be reclassified as definite SS. According to these data, the management of the disease should not be delayed by the absence of a definite diagnosis as this may be present in less than half of patients. More recently, it was proposed that there are two pathognomonic imaging findings that make the diagnosis of SS definite even when the clinical triad is not fulfilled: (1) the appearance of central callosal lesions on MRI and (2) arteriolar wall hyperfluorescence on retinal FA in normal appearing retinal arterioles that are far removed from any BRAO [23]. As these detailed descriptions were not provided in the medical records of our patients, these could not be applied to this case series.

Because of the rarity of SS, randomized controlled trials and prospective treatment studies for SS do not exist. Treatment has been based on the hypothesis of an autoimmune inflammatory endotheliopathic aetiology and on the reports of good response to immunosuppressive and immunomodulatory therapy in single-case reports and case series [11,24]. Recently, treatment guidelines have been proposed based on cumulative clinical experience and a large cohort of patients followed longitudinally, taking into account each organ involvement and the diverse severity of the disease [25]. In our cohort, a combination of glucocorticoids with other immunosuppressive agent and IVIG was the most commonly used strategy. However, given the design of the study, the variable response to therapy and its unstandardized monitoring plan, no further conclusions can be made on this issue.

The limitations of the study include its small size and retrospective design with resulting missing or undetailed data.

## 5. Conclusions

This series of nine patients with SS in a single centre highlights the variability in clinical presentations and should raise awareness to the early diagnosis and treatment, factors that play an important role in the prognosis of the patient. Future multicentre prospective studies are needed for better understanding of the syndrome.

## Figures and Tables

**Figure 1 jcm-11-06549-f001:**
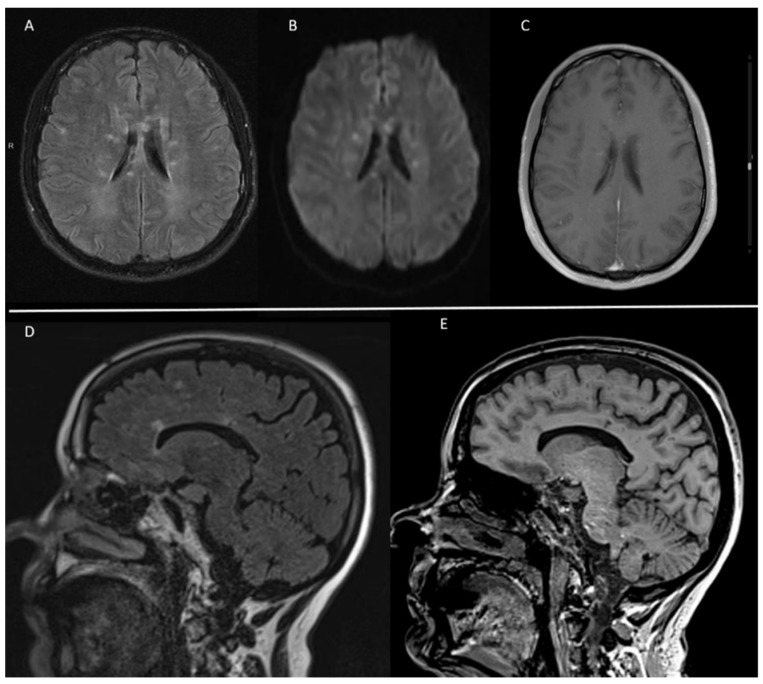
(**A**) Axial images of brain MRI on FLAIR sequence showing bilateral hyperintense lesions with preferential involvement of the corpus callosum. (**B**) Diffusion weighted sequence displaying bilateral acute small infarcts. (**C**) Some of the lesions presented contrast enhancement on T1 weighted sequence. (**D**,**E**) Sagittal images on FLAIR sequence and 3D-MPRAGE showing several characteristic “snowball” lesions in the corpus callosum.

**Figure 2 jcm-11-06549-f002:**
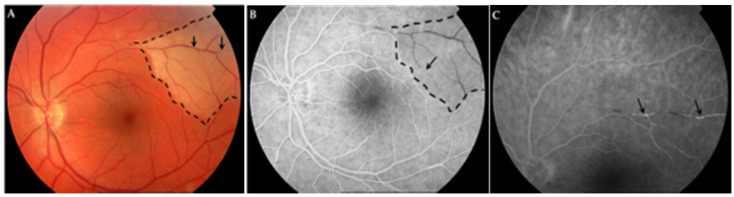
(**A**). Retinography: retinal edema in the upper temporal zone (dash line) with arterial occlusions (arrows). (**B**). Fluorescein angiography: area of retinal non-perfusion in the superior temporal zone (dash line) with segmental occlusions in arterial vessel (arrow). (**C**). Fluorescein angiography (late times): upper vessel hyperfluorescence reflecting inflammation (arrows).

**Table 1 jcm-11-06549-t001:** Demographic and clinical characteristics of nine patients with Susac syndrome.

	Case 1	Case 2	Case 3	Case 4	Case 5	Case 6	Case 7	Case 8	Case 9
**Sex**	Female	Male	Male	Male	Male	Female	Male	Female	Female
**Age at onset (years)**	19	27	23	46	33	30	59	54	37
**Age at diagnosis (years)**	20	27	23	46	34	41	59	54	38
**Diagnosis delay (months)**	12	5	1	0.5	8	120	5	2	1
**Organ involvement at disease onset**	Complete triad	CNS and eye	Complete triad	CNS and eye	CNS and eye	Ear	CNS and eye	CNS and ear	Complete triad
**Symptoms during evolution**	Additional retinal artery thrombosis, aggravated hearing loss, and new CNS lesions	Repeated alteration of consciousness and new CNS lesions	Symptoms resolved and no relapse	Recurrent neurological symptoms	Additional ear involvement	Additional retinal artery thrombosis and decreased right cervical muscle strength	Complete triad in five months and recurrent neurological and eye involvement	Recurrent neurological symptoms	Symptoms resolved and no relapse
**CNS manifestations**	Headache, Left facial paresthesia	Headache, Impaired consciousness	Behavioral changes, Impaired consciousness, Confusion/ disorientation	Headache, Cognitive impairment, Confusion, Behavioral changes, Facial paresthesia, Gait impairment	Cognitive impairment, Impaired consciousness, Confusion, Hands paresthesia, Gait impairment	Headache, Decreased right cervical muscle strength	Headache, Cognitive impairment, Confusion/disorientation, Diplopia	Headache, Cognitive impairment, Confusion/ Disorientation, Hemimotor-hemisensitive syndrome, Paresthesias, Dysarthria, Diplopia, Ataxia	Cognitive impairment, Confusion/disorientation, Right-sided hemiparesis, Gait ataxia, Dysarthria
**Eye manifestations**	Bilateral visual loss	Blurred vision, Scotoma	Scotoma	-	Photopsias Scotomas	Visual loss of the right eye, Scotoma	Visual loss, Hemianopsia, Blurred vision	-	Blurred vision
**Ear manifestations**	Bilateral hearing loss	-	Right hearing loss	-	Bilateral hearing loss	Left hearing loss, Vertigo, Tinnitus	Left hearing loss, Vertigo	Left hearing loss	Bilateral hearing loss, Vertigo
**Disease course**	Polycyclic	Not classifiable	Monocyclic	Monocyclic	Monocyclic	Polycyclic	Monocyclic	Polycyclic	Monocyclic
**Number of relapses**	4	3	0	2	0	1	3	3	0

Abbreviations: CNS: central nervous system.

**Table 2 jcm-11-06549-t002:** Presentation and disease course in the nine patients with Susac syndrome.

	Organ Involvement	At Disease Presentation or During Follow-up			Organ Damage at Last Visit		
Patient		CNS	Eye	Ear	CNS	Eye	Ear
1							
2							
3							
4							
5							
6							
7							
8							
9							
	Organ involvement at presentation Organ involvement during disease course Organ damage at last visit						



Abbreviations: CNS: central nervous system.

**Table 3 jcm-11-06549-t003:** Diagnostic workup, treatment, and outcome of nine patients with Susac syndrome.

	Case 1	Case 2	Case 3	Case 4	Case 5	Case 6	Case 7	Case 8	Case 9
**MRI**	Callosal, periventricular, supratentorial grey matter pontine	Callosal, periventricular	Callosal, periventricular, supratentorial grey matter, basal ganglia, cerebellum, brainstem, perivascular and leptomeningeal contrast uptake	Callosal periventricular, basal ganglia	Callosal	Callosal, periventricular, subcortical cerebellum	Callosal periventricular, subcortical	Callosal, periventricular, centrum semiovale pontine, necrotic lesion on corona radiata	Callosal, periventricular, subcortical, basal ganglia, internal capsule, cerebellum
**Visual acuity**	Reduced	Reduced	Reduced	Normal	Reduced	Reduced	Reduced	Normal	Normal
**Visual field**	Altered	Altered	Altered	Altered	Altered	Altered	Altered	Normal	Not available
**Retinal fluorescein angiography**	Bilateral BRAO	Unilateral BRAO with infarction and bilateral vasculitis	Unilateral BRAO ^a^, bilateral peripheral retinal non-perfusion	Bilateral BRAO with infarction	Bilateral peripheral retinal non-perfusion with unilateral vasculitis	Bilateral BRAO with infarction and vasculitis	Bilateral BRAO with infarction	Normal	BRAO with vasculitis
**Audiometry**	Bilateral sensorineural hearing loss	Normal	Right transmission hypoacusis	Normal	Bilateral sensorineural hearing loss	Left sensorineural hearing loss	Left sensorineural hearing loss	Bilateral sensorineural hearing loss	Bilateral sensorineural hearing loss
**Cerebrospinal fluid**	Pleocytosis	Pleocytosis, high levels of protein	Pleocytosis, high levels of protein	High levels of protein	-	Normal	High levels of protein	Normal	High levels of protein
**Immunologic profile** **ANA** **APL** **ANCA** **AECA**	1/80 Negative Negative Negative	1/80 aB2GPI IgM 43.3U/mL and weak LAC Negative NA	Negative Negative Negative Negative	Negative Negative Negative NA	Negative Negative Negative NA	1/40 Negative Negative NA	Hep2 3–4 Negative Negative NA	1/80 Negative Negative NA	Negative Negative Negative NA
**EEG**	NA	NA	Abnormal: diffuse slow background activity of frontal predominance, FIRDA type	Abnormal: generalized slowdown in bioelectric activity	NA	NA	NA	NA	NA
**Diagnostic criteria at diagnosis**	Definite SS	Probable SS	Probable SS	Probable SS	Definite SS	Definite SS	Probable SS	Probable SS	Definite SS
**Treatment ^b^**	IV MP, oral steroids, MMF, IVIG, Cyclo, infliximab, RTX, ASA, plasma exchange, autologous hematopoietic stem cell transplantation	IV MP, oral steroids, cyclo, IVIG, ASA	IV MP, oral steroids, cyclo, ASA	IV MP, oral steroids, IVIG, ASA	IV MP, oral steroids, cyclo, ASA	IV MP, oral steroids, cyclo, IVIG, OAC, ASA	IV MP, oral steroids, cyclo, IVIG, MMF, ASA	IV MP, oral steroids, cyclo, IVIG, RTX, MMF, ASA	Oral steroids, IVIG, MMF, ASA
**Residual damage**	Headache, hemiparesis, visual deficit, and bilateral hearing loss	None	Hearing loss, visual deficit	Headache, instability	Hearing, visual and memory impairment	Hearing and visual deficit with glaucoma	None	Cognitive impairment, left lower extremity monoparesis and left hearing loss	Hearing impairment
**Final Rankin**	1	0	1	1	1	1	0	2	3
**Follow-up (months)**	22	6	71	44	113	131	38	35	56

Abbreviations: aB2GPI: anti-β2 glicoprotein I antibody; AECA: anti-endothelial cell antibodies; ANA: anti-nuclear antibodies; ANCA: anti-neutrophil cytoplasm antibodies; APL: anti-phospholipid antibodies; ASA: aspirin; BRAO: branch retinal artery occlusion; CNS: central nervous system; CSF: cerebrospinal fluid; cyclo: cyclophosphamide; EEG: electroencephalogram; FIRDA: frontal intermittent rhythmic delta activity; IVIG: intravenous immunoglobulin; LAC: lupus anticoagulant; MMF: mycophenolate mofetil; IV MP: intravenous methylprednisolone; MRI: magnetic resonance imaging technique; NA: not available; OAC: oral anticoagulation; RTX: rituximab. ^a^ Data obtained from ocular funduscopy and optical coherence tomography angiography (OCT-A); retinal fluorescein angiography was not performed. ^b^ Treatments are presented in order of administration.

**Table 4 jcm-11-06549-t004:** Comparison of patient characteristics of the present series with previous published SS series or cohorts.

	Present Series	Dorr et al., 2013 [7]	Jarius et al., 2014 [3]	Karahan et al., 2019 [16]	Triplett et al., 2022 [17]
Number of patients	9	304	25	19	32
Sex (%)					
Female	44	78	72	63	53
Male	55	22	28	37	47
Age at onset (years)					
Range	19–59	8–65	17–56	19–66	21–61
Mean or median	36	32	28	33	37
Symptoms at onset (%)					
Complete triad	33	13	0	16	19
CNS involvement	88	67	72	95	100
Eye involvement	78	40	24	42	38
Ear involvement	56	37	20	42	63
Manifestations during disease course (%)				NA	NA
Complete triad	66	85	NA		
CNS involvement	100	91	88		
Eye involvement	89	97	96		
Ear involvement	78	96	96		
Duration of symptoms prior to diagnosis (months)					
Range	0.5–120	NA	0–126	0–39	0.5–100
Mean or median	5	5	7	3	3
Length of follow-up (months)					NA
Range	5–131	1–252	0–204	0–84	
Mean or median	44	41	54	36	
Relapsing course					NA
% of patients	33	42	76	16	
Range of relapses	1–4	1–10	1–6	0–1	

Abbreviations: CNS: central nervous involvement; NA: not available.

## Data Availability

The datasets generated and/or analyzed during the current study are available from the corresponding author on reasonable request.

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
