# Peer review of "Susac Syndrome: Description of a Single-Centre Case Series"

_jcm, 2022, doi:10.3390/jcm11216549_

Round 1

Reviewer 1 Report

Beca and colleagues present the clinical characteristics of a cohort of 9 patients diagnosed with Susac’s syndrome (SuS) between 2006 and 2020. The cohort differs from other published cohorts in having a slight male predominance but otherwise shares many characteristics with other published cohorts. The article highlights that patients often present without the full diagnostic clinical triad at presentation leading to delays in diagnosis. It is well written and contains important and relevant information about Susac’s syndrome. It is critical that even relatively small cohorts of SuS are published as detailed information about the condition is important to add to the literature.

Comments to the Author

1.     The authors include information relating to different “forms” of SuS including monophasic, chronic continuous and chronic relapsing forms. This classification of SuS is controversial e.g. Vodopivec and Prasad, 2017. I think the authors need to include a discussion of the difficulty in trying to classify SuS into these categories when patients have short follow-up times, and may be given different treatments which might arrest the disease in some, and undertreat the disease in others. It may be that SuS is a single disease with a spectrum of outcomes but not necessarily with different forms.

2.     Please avoid using the word gender when talking about biological sex.

3.     I would suggest citing Hardy et al. 2015 JNNP in the Intro when listing publications indicating that SuS appears to be T-cell mediated as this was an important publication on this topic.

Author Response

REVIEWER #1

Beca and colleagues present the clinical characteristics of a cohort of 9 patients diagnosed with Susac’s syndrome (SuS) between 2006 and 2020. The cohort differs from other published cohorts in having a slight male predominance but otherwise shares many characteristics with other published cohorts. The article highlights that patients often present without the full diagnostic clinical triad at presentation leading to delays in diagnosis. It is well written and contains important and relevant information about Susac’s syndrome. It is critical that even relatively small cohorts of SuS are published as detailed information about the condition is important to add to the literature.

  • The authors include information relating to different “forms” of SuS including monophasic, chronic continuous and chronic relapsing forms. This classification of SuS is controversial e.g., Vodopivec and Prasad, 2017. I think the authors need to include a discussion of the difficulty in trying to classify SuS into these categories when patients have short follow-up times and may be given different treatments which might arrest the disease in some and undertreat the disease in others. It may be that SuS is a single disease with a spectrum of outcomes but not necessarily with different forms.

We appreciate the opportunity to include the following comment on this matter in our manuscript (Discussion section).

Although the clinical course of SS has been broadly characterized as monocyclic, polycyclic, or chronic continuous, the validity of these concepts was assessed by Vodopivec I. and Prasad S. Neuroophthalmol. 2017, and they found that a short follow-up bias in reported cases confounds the true long-term outcome of the disease, considering that relapses have been observed after 10 years of remission. Importantly, the distinction between a relapse and unremitting low-grade disease activity may be challenging, given the lack of accepted definitions for relapse, remission, or intermittent disease. Therefore, the stratification of SS into monocyclic, polycyclic, or chronic continuous forms may be inaccurate, and SS may be best regarded as a disease with a wide range of outcomes that may be altered by treatment.

  • Please avoid using the word gender when talking about biological sex.

We thank the reviewer for the suggestion and agree with it. The corrections have been made.

  • I would suggest citing Hardy et al. 2015 JNNP in the Intro when listing publications indicating that SuS appears to be T-cell mediated as this was an important publication on this topic.

Hardy el al JNNP 2015 described the brain histopathology in three cases of SS, reporting microinfarcts and T-cell inflammation involving small-sized to medium-sized vessels. Considering the important contribution for the better understanding of the pathogenesis of this disease, we appreciate the suggestion of the reviewer, and added this reference to the Introduction section of the manuscript.

Reviewer 2 Report

Summary:

-This retrospective case series of patients with Susac syndrome does have some interesting findings, but these aren’t particularly well-articulated in the manuscript. 

-The English quality and grammar requires improvement throughout.

-I think the manuscript could be improved by better presentation of the data using a figure or 2 instead of long tables. Instead a table comparing key features of this cohort could be compared with previously published cohorts to better highlight the similarities and differences.

Results:

-I think some of the results would be better represented using figures e.g. table 3 could be a Venn diagram demonstrating the overlapping clinical syndromes.

-A figure demonstrating some of the imaging and fluroscein angiography findings might be of interest to readers.

Discussion:

-No mention is made of the use of cocaine in a few of the included patients prior to presentation. Is cocaine use a precipitant or trigger to the clinical syndrome in susceptible individuals? Was cocaine use only in the monocyclic patients, in which case was this drug-induced vasculopathy? I think this requires discussion.

Author Response

REVIEWER #2:

Summary:

-This retrospective case series of patients with Susac syndrome does have some interesting findings, but these aren’t particularly well-articulated in the manuscript.

- The English quality and grammar require improvement throughout

1) I think the manuscript could be improved by better presentation of the data using a figure or 2 instead of long tables. Instead, a table comparing key features of this cohort could be compared with previously published cohorts to better highlight the similarities and differences.

To address this topic, we reformulated the table 1 and particularly the table 2, in order to uniformize and simplify the data, reducing considerably its length and enhancing its clarity for benefit of the reader. Although we appreciated the reviewer´s comment, as a description of a case series with less than 10 patients, we considered useful and illustrative to describe each case on the manuscript. For this reason, we maintained its initial structure with the referred adjustments for your consideration. It would also be possible to convert table 1 and 2 (now renamed 3) into supplementary material, if that is considered the best option by the Editor and reviewers. An additional table (table 4) was created, comparing our series with previously published ones, as suggested by the reviewer. It was included in the Discussion section.

2) I think some of the results would be better represented using figures e.g., table 3 could be a Venn diagram demonstrating the overlapping clinical syndromes.

As suggested, we changed the table 3 (now renamed table 2) to a more visual version, which hopefully demonstrates more clearly the overlapping of the three organ involvements and the corresponding moment of presentation.

3) A figure demonstrating some of the imaging and fluroscein angiography findings might be of interest to readers.

Thank you for the suggestion. We added a figure demonstrating the central nervous system findings on brain magnetic resonance imaging, and another with the most representative ocular findings on fluorescein angiography.

4) Discussion: No mention is made of the use of cocaine in a few of the included patients prior to presentation. Is cocaine use a precipitant or trigger to the clinical syndrome in susceptible individuals? Was cocaine use only in the monocyclic patients, in which case was this drug-induced vasculopathy? I think this requires discussion.

In our series, three patients had history of cocaine consumption. In this context, the rare diagnosis of multifocal inflammatory leukoencephalopathy or a toxic-induced vasculopathy should be considered. When these patients were compared with those without cocaine exposure, there were no differences regarding the clinical manifestations, results of diagnostic workup or response to therapy, thus suggesting that the diagnosis of SS was accurate. Anecdotally, these patients had the shorter time from symptoms onset to diagnosis, and two had behavioral changes, not reported in the rest of cases. All had a monocyclic course of the disease, similar to the majority of our series, but limited information was detailed on medical records concerning the pattern of consumption or abstinence during follow-up. On another perspective, the potential contribution of toxic agents to the development of SS has not been extensively investigated, although some authors (Hantson et al J Med Toxicol. 2015, De Baerdemaeker et al Case Rep Neurol. 2020), reporting single cases, have suggested that cocaine use, particularly in the presence of adulteration by levamisole, could play a role as a triggering cofactor in the onset of SS or Susac-like syndromes. However, a causal association between drug use and the appearance of SS cannot be established.

We added this information as a paragraph in the Discussion section.

Round 2

Reviewer 1 Report

The authors have addressed my comments

Reviewer 2 Report

Thank you for addressing my comments. I have no further comments to add.